

# Avoidance of copper by fathead minnows (*Pimephales promelas*) requires an intact olfactory system

Rubab Fatima[1], Robert Briggs[1] and William A. Dew[1,2]

[1] Biology, Trent University, Peterborough, Ontario, Canada
[2] Biology, Algoma University, Sault Ste. Marie, Ontario, Canada

## ABSTRACT

Fish can detect and respond to a wide variety of cations in their environment, including copper. Most often fish will avoid copper during behavioural trials; however, fish may also show no response or an attraction response, depending on the concentration(s) used. While it may seem intuitive that the response to copper requires olfaction, there is little direct evidence to support this, and what evidence there is remains incomplete. In order to test if olfaction is required for avoidance of copper by fathead minnows (*Pimephales promelas*) copper-induced movement was compared between fish with an intact olfactory system and fish with induced anosmia. Fish in a control group or a mock-anosmic group avoided copper (approximately 10 μg/L or 62.7 nM copper sulphate) while anosmic fish did not. The evidence demonstrates that an intact olfactory system is required for copper sensing in fish.

## INTRODUCTION

The olfactory system in fish is able to recognize a wide array of chemical cues within the environment, including cues to find food, identify kin, evade predators, and mating (*Zielinski & Hara, 2006*; *Tierney et al., 2010*; *Tierney, 2016*). Behavioural responses of fish to cations has been shown to a variety of cations including, but not limited to, calcium (*Bodznick, 1978*; *Dew & Pyle, 2014*; *Poulsen et al., 2014*), zinc (*Sprague, 1964*), nickel (*Giattina, Garton & Stevens, 1982*), cadmium (*Giattina, Garton & Stevens, 1982*), chromium (*Hartwell et al., 1989*; *Svecevičius, 2007*), arsenic (*Hartwell et al., 1989*), and cobalt (*Hansen et al., 1999a*), as well as complex metal mixtures (*Sprague, 1964*; *Hansen et al., 1999a*). The most studied cation in terms of behavioural responses is copper.

Copper induces avoidance behaviours in a wide range of species including the ten-spined stickleback (*Pygosteus pungitius*) (*Jones, 1947*), goldfish (*Carassius auratus*) (*Kleerekoper et al., 1972*; *Westlake, Kleerekoper & Matis, 1974*), golden shiners (*Notemigonus crysoleucas*) (*Hartwell et al., 1989*), smelt (*Retropinna retopinna*) (*Richardson, Williams & Hickey, 2001*), inanga (*Galazias maculatus*) (*Richardson, Williams & Hickey, 2001*), the common bully (*Gobiomorphus cotidianus*) (*Richardson,*

Corresponding author
William A. Dew,
william.dew@algomau.ca

*Williams & Hickey, 2001*), and a variety of salmonids (*Sprague, 1964*; *Hansen et al., 1999b*; *Sommers et al., 2016*; *Van Genderen et al., 2016*). While most studies show avoidance of copper, some also demonstrate preference and no response. Collectively the papers cited above demonstrate avoidance to copper at a wide range of concentrations, with preference for copper only occurring at the highest concentrations tested. For example, rainbow trout (*Oncorhynchus mykiss*) avoid copper at a concentration less than 300 µg/L and seem to prefer waters containing 330–390 µg/L (*Giattina, Garton & Stevens, 1982*), while lake whitefish (*Coregonus clupeaformis*) avoid copper at concentrations of 100–5,000 µg/L and spend more time in waters containing copper in excess of 10,000 µg/L copper (*Hara, 1981*). While most attraction responses occur to high concentrations of copper, attraction to copper has been demonstrated at a concentration as low as 11 µg/L in goldfish (*Kleerekoper et al., 1972*).

For all the work that has been done to determine how fish respond to copper, very little has been done to demonstrate how fish detect copper in their environment. Neurophysiological measures of the olfactory system, namely electro-encephalograms (EEGs) at the olfactory bulb and electro-olfactograms (EOGs) at the olfactory epithelium have shown that the olfactory system in fish responds to calcium, sodium, and magnesium, but not potassium (*Bodznick, 1978*; *Hubbard, Barata & Canario, 2000*; *Dew & Pyle, 2014*). A recent study has also demonstrated that rainbow trout show an EOG response when presented with copper, cadmium, or nickel as an olfactory cue (*Lari, Bogart & Pyle, 2018*). Only one study to date has investigated the responses of the olfactory system to copper *via* direct neurophysiological measures and behavioural responses, and they concluded that lake whitefish show a depression in background EEG activity to increasing concentrations of copper, which is not a response normally associated with perception of an odour (*Hara, 1981*). The apparent disconnect between the behavioural response and the direct neurophysiological measurements in lake whitefish raises the question of whether the response of fish to copper is mediated by olfaction or another sensory modality.

In this study fathead minnows (*Pimephales promelas*) were used to test if the avoidance response of a cyprinid to copper is dependent on olfaction. The behavioural response of fathead minnows to copper was compared between intact fish, anosmic fish, and fish that had undergone a mock-anosmic treatment. We hypothesize that fathead minnows will avoid copper when olfaction is intact and will not avoid copper when anosmic. This study design mirrors a study previously performed with calcium that demonstrated calcium is an odorant for fathead minnows (*Dew & Pyle, 2014*).

## MATERIALS AND METHODS

Adult fathead minnows were obtained from Laurier University's Center for Cold Region and Water Science facility from a breeding colony and housed at the aquatic facility at Trent University. Minnows were held in static tanks filled with ozonated and sand-filtered Otonabee river water held at 20 °C with supplemental aeration. All water was allowed to age for 24 h to allow excess ozone and oxygen to off-gas to ensure oxygen concentrations were appropriate for use with fish. A 12 h:12 h light:dark cycle was maintained in the holding room and fish were fed twice a day on a diet of live brine shrimp and fish flake.

Enrichment was provided in the form of spawning tiles and other structures. Prior to being used for any experiments, fish acclimated to lab conditions for 2 weeks. Fish were monitored daily for disease, distress, or an inability to right themselves (*i.e.*, maintain balance), all of which were criteria for euthanasia (no animals were euthanized in the holding tanks). All procedures were approved by the Trent University Animal Care Committee (IACUC number 24420) in accordance with guidelines from the Canadian Council on Animal Care.

All solutions, including water used for all behavioural trials, were made using filtered and ozonated Otonabee River water (*i.e.*, the same source as the holding water). Behavioural trials were conducted using I-mazes (55 cm × 11.5 cm × 14 cm; length × width × height, respectively) that had a piece of white corrugated plastic (coroplast) affixed to the bottom of the maze sectioned into three equally sized zones. Each maze was filled with 3 L of Otonabee River water and movement of fish in the chamber was monitored using a webcam suspended above the behavioural maze. A copper sulphate solution (1.5 mg/L or 9.4 mM) was prepared in Otonabee River water so that when a 10 mL aliquot was administered at one end of the behavioural maze it would have diffused to make one half of the water, a volume of 1.5 L, a concentration of approximately 10 μg/L (62.7 nM). One fish was randomly selected from the appropriate holding tank and placed into a maze. A white curtain was then drawn around the mazes to prevent any movement by the researchers from affecting fish behavior. The fish were allowed to acclimate for 5 min. After the acclimation period 10 mL of the copper solution was administered to one randomly selected (*via* coin flip) end of the maze (*i.e.*, the copper-containing end) and 10 mL of aged ozonated Otonabee river water was administered to the other end (*i.e.*, the blank end). Preliminary dye trials demonstrated that it took 3 min for the dye to reach the center of the maze, therefore the copper and blank solutions were allowed 3 min to diffuse after being added. After the diffusion period, the position of the fish in the maze was then recorded every 10 s for 5 min. Five mazes were monitored at the same time, with each maze being washed and thoroughly rinsed after each trial to ensure no carry-over between trials. Preliminary trials were performed to measure the response of intact fish taken from the holding tank to copper. All experimental groups (control, mock anosmic and anosmic) were interspersed so that at least one fish per group were done in a group of 5. No fish demonstrated a typical fright response of remaining still during the study. The identity of the group a fish belonged to was known to researchers at all times, but given the nature of the measurements (instance in a section of the maze), it is unlikely this influenced results.

Anosmia was induced using the technique described in *Dew & Pyle (2014)*. Fish were anesthetized in 1 L of aged ozonated Otonabee River water containing 120 mg of tricaine methanesulfonate, or MS-222, (anesthetic agent) and 240 mg of sodium bicarbonate (NaHCO$_3$) to bring the final solution to pH 7.0. Once the fish began to lose its ability to swim upright in the tank, it was removed from the tank, each naris cleared of water using a small piece of paper towel, and a small amount of tissue glue (1–2 μL) (3M Vetbond, Milton, ON) was inserted into each nasal cavity using a micropipette. This was just enough tissue glue to cover the olfactory epithelium without covering any other structures such as the eyes or gills. This procedure was done as quickly and efficiently as possible with the fish
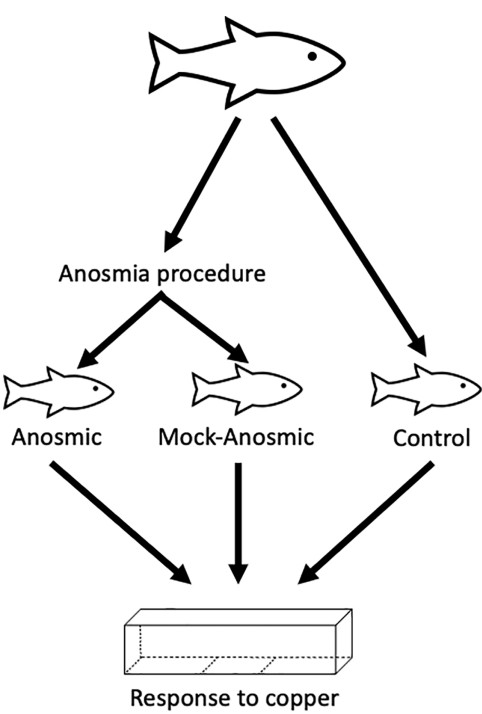

**Figure 1 Schematic of experimental design.** There were three groups of fish used for this experiment, fish taken directly from holding, fish made anosmic, or fish that went through the anosmia-inducing procedure but were not made anosmic (*i.e.*, mock-anosmic). The response of each group of fish to copper was assessed using a trough maze.

being placed into a labeled recovery tank. A mock anosmic treatment was performed on a second group of fish in which fish were handled in the same manner as the anosmic group (*i.e.*, anesthetized with MS-222), with 1–2 μL of water added to each olfactory chamber instead of tissue glue. The only difference between the anosmic and mock anosmic treatments was the presence of tissue glue in the anosmic group. All fish that underwent anesthesia were then allowed to recover in a tank of clean water for a minimum of 24 h. The behavioural method described above was performed on anosmic and mock anosmic fish, as well as a control group consisting of fish taken directly from a holding tank. An $n = 14$ per group (the experimental unit was an individual fish) was used for the behavioural trials, which is supported by a previous study using calcium as an avoidance cue (*Dew & Pyle, 2014*). All fish in the anosmic group were euthanized with an overdose of MS-222 at the end of the experiment, all other fish were transferred to other protocols. A schematic detailing the experimental design is found in Fig. 1.

The proportion of time spent in the copper-containing arm was calculated for each fish as per Eq. (1) (*Lari et al., 2015*; *Fischer & Dew, 2021*):

$$Preference = \frac{(time\ in\ cue\ arm\ -\ time\ in\ control\ arm)}{(time\ in\ cue\ arm\ +\ time\ in\ control\ arm)} \tag{1}$$

A value at or near 0 indicates fish spent an equal amount of time in either end, a value significantly less than 0 would indicate that fish spent less time in the copper-containing

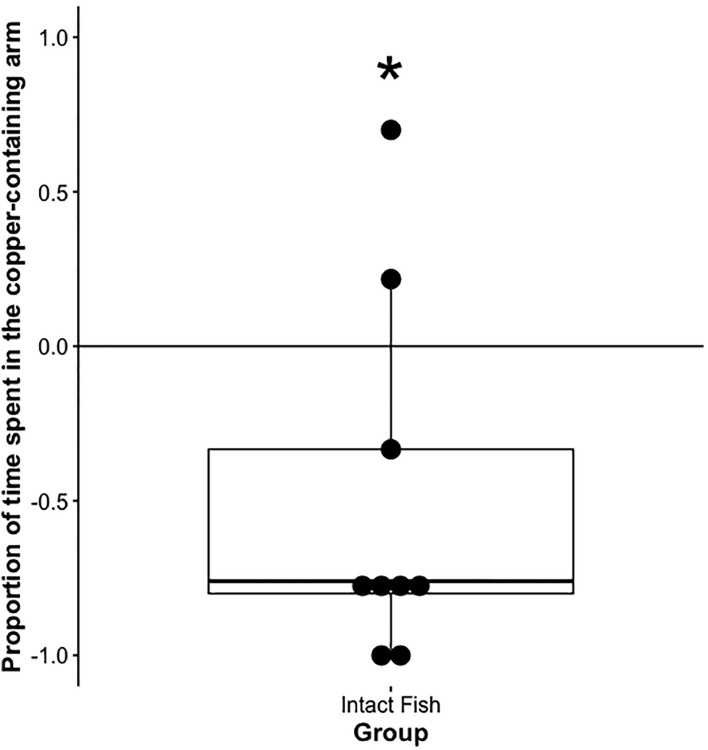

**Figure 2 Average proportion of time spent in the copper-containing arm—preliminary study.** The mean +/− SEM for the proportion of time spent in the copper-containing arm of the maze (*i.e.*, value for copper arm preference) for animals removed from the holding tank. An asterisk (*) denotes significant difference from 0, *n* = 9.

arm than the blank arm (*i.e.*, they avoided the cue), and a value above 0 indicates attraction to the cue. Parametric assumptions held for each trial and a one-sample t-test against 0 was performed for each of the three treatments to determine if any treatment resulted in more time being spent in either end of the maze. A Holm-Bonferroni sequential correction was applied to produce corrected *p*-values due to the use of multiple t-tests, with a *p*-value of <0.05 being considered significant. Cohen's d was calculated for effect size using the effsize package (*Torchiano, 2020*). An ANOVA followed by a Tukey's Honest Significant Difference test as a post-hoc was performed to compare among and between the different groups. An $\eta^2$ value was calculated for the effect size for the ANOVA using the lsr package in R (*Navarro, 2015*). All analyses were done using R (*R Core Development Team, 2021*) with graphs produced in R using the ggpubr package (*Kassambara, 2020*). Raw data have been provided as Supplemental Material.

## RESULTS

In preliminary trials, fish taken from the holding tanks avoided copper (t(8) = −2.55, *p* = 0.034, *n* = 9, df = 8, d = −0.85; Fig. 2). For the experimental groups, fish avoided the copper containing arm in the control (t(13) = −3.66, *p* = 0.006, *n* = 14, df = 13, d = −0.97; Fig. 3) and mock anosmic (t = −5.71, *p* < 0.001, *n* = 14, df = 13, d = −1.52) treatment groups, but not in the anosmic group (t(13) = 1.23, *p* = 0.24, *n* = 14, df = 13, d = 0.33).
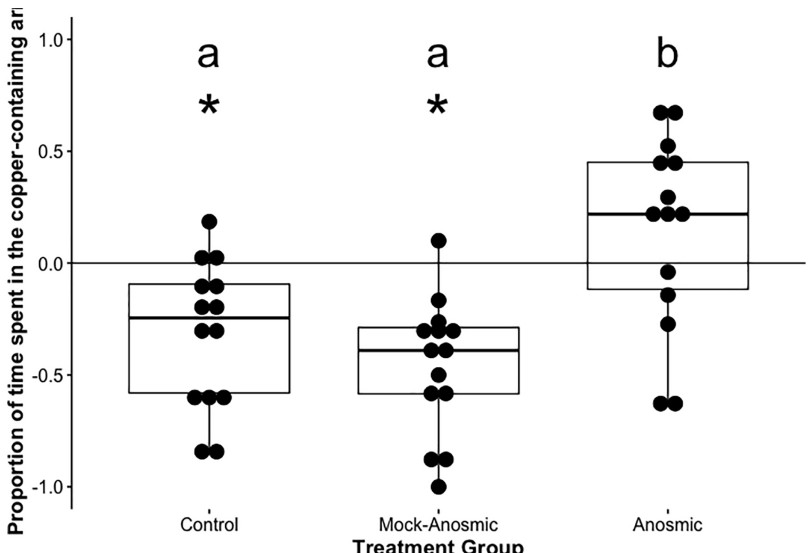

**Figure 3 Average proportion of time spent in the copper-containing arm—anosmia trials.** The mean +/− SEM for the proportion of time spent in the copper-containing arm of the maze (*i.e.*, value for copper arm preference). An asterisk (*) denotes significant difference from 0, dissimilar letters denote a significant difference, $n = 14$ for each trial. 

Direct comparison of the groups showed that the mock anomsic and control groups were not significantly different; however, the anosmic group was significantly different from both the mock anosmic and control group ($F_{(2,39)} = 1.78$, $p < 0.001$, $n = 42$ (14 per group), df = (2,39), $\eta^2 = 0.36$).

## DISCUSSION

The data demonstrate that without an intact olfactory system, fathead minnows do not avoid copper in the environment. This is supported by a study that used chemical ablation of the olfactory epithelium with nitric acid to demonstrate impairment of some, but not all, behavioural responses to copper (*Hara, 1981*). There is lack of clarity in the literature concerning the response of fish to copper as measured using neurophysiological methods. Rainbow trout show an EOG response to copper while whitefish show a decrease in background electrical activity levels due to increasing copper concentration, something not normally associated with an olfactory response (*Hara, 1981*; *Lari, Bogart & Pyle, 2018*). The apparent discrepancy between the EOG and EEG results could be due to several reasons. An EOG measures the bulk response of olfactory sensory neurons (*i.e.*, EOG measures the generator potential of many neurons), while EEG measures electrical activity at the olfactory bulb. It may be that olfactory sensory neurons are responding but the signal is not being propagated to the olfactory bulb, or it could be that a depression in electrical activity is due to impairment of the olfactory system by copper. In the *Hara (1981)* study they used concentrations of copper from 25–2.5 mg/L, which is greater than what has been demonstrated to impair olfaction in short-term exposures (*Green et al., 2010*; *Dew, Wood & Pyle, 2012*). It is possible, then, that when the copper was perfused over the olfactory epithelium in the *Hara (1981)* study there was an immediate impairment of the olfactory

system that prevented any response to the cue. That being said, it is unclear why EOG responses would not have been impaired in rainbow trout as both studies used similar concentrations. Despite a possible impairment of olfaction as measured by EEG, whitefish demonstrated an avoidance to low concentrations of copper. This avoidance was most likely due to the fact they were exposed to a concentration gradient of copper in the behavioral maze, meaning they were likely avoiding copper and concentrations far below what was used for the EEG experiments. It is also possible that the response to copper as measured by EEG, EOG, or behaviour is not due to copper being an odorant, but instead as a response to damage to the olfactory epithelium caused by copper. Regardless of the reason for the discrepancy, our study clearly demonstrates that avoidance of copper requires an intact olfactory system, regardless of questions raised by the neurophysiological results of previous studies.

A variety of reasons have been proposed as to why fish respond to copper in their environment. The attraction response seen at low concentrations of copper in goldfish may be due to the fact that copper is an essential element for fish and that by moving into an area with a low concentration of copper the fish can absorb needed copper through the gills (*Kleerekoper et al., 1972*). At higher concentrations copper can cause a variety of toxic endpoints in fish, therefore avoidance of areas with higher concentrations of copper may be an adaptive response to prevent toxicity. A bigger question is why do fish demonstrate an attraction response to copper at concentrations well above those demonstrated to show olfactory impairment? It has been suggested that when fish swim into an area of high copper they become stupefied and unable to swim out of the copper due to a toxic effect, therefore, the preference seen at high concentrations may not truly be a preference for copper but the result of a toxic endpoint (*Jones, 1947*; *Kleerekoper et al., 1972*). Fish may also avoid areas with a differing concentration of cations to avoid osmotic pressure that would result from moving into an area with a drastically different cation concentration. Regardless of why fish respond to copper, this study demonstrates that without an intact olfactory system, there is no behavioural response.

## ACKNOWLEDGEMENTS

The authors wish to thank Deborah MacLatchy and Andrea Lister from Wilfred Laurier University for providing the fathead minnows used in this study. The authors would also like to thank Jason Allen and the animal care staff at the Trent University Aquatic Facility for their help during this study. The coauthors declare no conflict of interest.

### Funding

The authors received no funding for this work.

### Competing Interests

The authors declare that they have no competing interests.

## Author Contributions

- Rubab Fatima performed the experiments, analyzed the data, authored or reviewed drafts of the article, and approved the final draft.
- Robert Briggs performed the experiments, analyzed the data, authored or reviewed drafts of the article, and approved the final draft.
- William A. Dew conceived and designed the experiments, analyzed the data, prepared figures and/or tables, authored or reviewed drafts of the article, and approved the final draft.

## Animal Ethics

The following information was supplied relating to ethical approvals (*i.e.*, approving body and any reference numbers):

Trent University Animal Care Committee.

## Data Availability

The raw data is available in the Supplemental File.

## Supplemental Information

Supplemental information for this article can be found online at http://dx.doi.org/10.7717/peerj.13988#supplemental-information.

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
