# Peer review of "Avoidance of copper by fathead minnows (Pimephales promelas) requires an intact olfactory system"

_PeerJ, doi:10.7717/peerj.13988_

## Round 0.1 · original submission · Major Revisions

Dear authors, please kindly address the comments and concerns of each reviewer. In addition, my comments are as below:

1. The experimental design to investigate the behavioural response of fathead minnows towards copper is interesting. However, as pointed out by one of the reviewers, this is only a single behavioural experiment, with one measurable variable (preference based on time spent) being used for comparison purpose. This study, albeit addresses an important issue in fish behaviour, could benefit from the inclusion of additional data to support the conclusion. For example, does the avoidance of copper behaviour differ between male and female adult fish? How about fish of different body size? Sex and body size were not described in the study.

2. The inclusion of a schematic diagram or figure outlining the experimental design is needed to aid readers to understand the whole experiment.

3. There are minor language issue that needs to be addressed.

4. Please highlight what is the difference between this research and that of Dew et al. (2012) (https://dx.doi.org/10.1021/es300670p). Dew et al. tested the effect on the olfactory of fathead minnows exposed to varying concentrations and exposure durations of copper.

Reviewer 1 ·

Excellent Review

This review has been rated excellent by staff (in the top 15% of reviews)
EDITOR COMMENT
This review provides a clear overview of the reviewed manuscript, and constructive comments and suggestions for the authors to improve upon. Additionally, the reviewer even went to the extra mile to double-check the results using the raw data provided. The positive and polite tone throughout the reviewer's comments further shows his/her professionalism in reviewing this manuscript.

Basic reporting

This manuscript describes an experiment wherein minnows were exposed to a sublethal concentration of copper sulphate and their behavioural response in terms of time spent in either end of the tank (i.e., avoidance/attraction/indifference) recorded. The authors found that untreated and sham-anosmic fish tended to avoid the end of the tank to which the copper had been added, whereas anosmic fish were indifferent. The authors conclude that minnows are able to detect copper via their olfactory system and thereby avoid it. I believe that the experimental design is simple, but adequate, and the results are clear. However, I have a slight issue with the conclusions drawn from this experiment; this point is discussed in the section 'Validity of the Findings'. Here, I list some minor comments.

Abstract
Lines 17-18: 'In order to demonstrate…' implies that you have already decided the outcome of your experiment; I suggest 'In order to test whether…'
Lines 20-21: Although the behavioural response of anosmic fish was different from that of controls, your experiment did not address detection of copper per se. I therefore suggest 'Control and sham-anosmic fish avoided copper whereas anosmic fish did not' or something similar.

Introduction
Line 68: I suggest 'odorant' rather than 'odour'; 'odour' is the characteristic smell of something and may be a complex mixture of many different individual odorants.
Materials and Methods
Lines 70-71: I think that it would be helpful if you clarified whether the minnows were bred in captivity or were caught in the wild.
Line 82: Please define exactly what you mean by 'aged water'.
Line 87: lease clarify whether 1.5 mg.l-1 refers to copper or copper sulphate. My preferred solution would be to express concentration as molar (e.g., 0.157 M)
Lines 102-104: I appreciate the inclusion of this definition of 'fright response' in the context of your experiment. However, as no fish showed such a response, perhaps it is sufficient to say 'no fish showed a typical fright response of remaining still'?
Line 108: It is probably best to define MS222.

Experimental design

The experimental design is simple and robust, and has been previously used in related studies. I have a minor suggestion and a question:

Lines 119-120: You say that fish were allowed to recover from anaesthesia for a minimum of 24 hours before experiment. I wonder what the maximum period was? Do you know how long minnows remain anosmic after such treatment? Can they recover their sense of smell?

The data have been provided as an Excel file, and the differences among treatments are clear and robust. However, I wonder whether a one-way ANOVA with Holm-Sidak post hoc would be a better method of statistical analysis rather than repeated t-tests? I appreciate that you wish to test whether fish avoided or were attracted to the copper end in each treatment, but these treatments form part of the same experiment. This may be a question borne of statistical ignorance; the result would be the same (I checked).

Validity of the findings

As you will be well aware, copper is toxic and can affect olfactory sensitivity of fish at concentrations way below lethal levels (for example, Baldwin et al., 2011; Hansen et al., 1999; Saucier et al., 1991), including the estimated level of your experiment (10 g.l-1 or 0.16 M). I think that this could - at least partially - explain the avoidance behaviour of the minnow; it could be that the fish are avoiding something that is causing damage to their olfactory system, rather than responding to specific detection of copper as an odorant. This toxicity could, as you suggest, explain the reduction in olfactory bulbar activity seen Hara (1981; I was unable to access this reference). It could also explain your minnows' avoidance behaviour. I believe that you should at least allow for this possibility in your discussion.

Baldwin, D.H., C.P. Tatara, and N.L. Scholz. 2011. Copper-induced olfactory toxicity in salmon and steelhead: Extrapolation across species and rearing environments. Aquat. Toxicol. 101: 295-297.

Hansen, J.A., J.D. Rose, R.A. Jenkins, K.G. Gerow, and H.L. Bergman. 1999. Chinook salmon (Oncorhynchus tshawytscha) and rainbow trout (Oncorhynchus mykiss) exposed to copper: Neurophysiological and histological effects on the olfactory system. Environ. Toxicol. Chem. 18: 1979-1991.

Saucier, D., L. Astic, and P. Rioux. 1991. The effects of early chronic exposure to sublethal copper on the olfactory discrimination of rainbow trout, Oncorhynchus mykiss). Environ. Biol. Fishes 30: 345-351.

Reviewer 2 ·

Basic reporting

This study designed to investigate that fathead minnows require an intact olfactory system to avoid copper exposure. This study is interesting and important as highlighted in the manuscript to fill up the missing knowledge from previous studies. This study found that fathead minnows in control and mock-anosmic groups were able to show avoidance behavior, but anosmic group was not. Thus, this study proven that fish (fathead minnows) require olfactory system to sense copper contamination. Thereby indicated the need of this study and the novelty of new knowledge can be generated from this study. References given were sufficient to highlight the context of the study scope. Nevertheless, results section was too brief and only ONE graph available. Perhaps there are some data available that 'mentioned used for other protocols' can be included here. Discussion section, should improve to discuss more about present result obtained especially highlight the mechanism of iono-sensory ogran in related with copper exposure. (detail refer attachment)

Experimental design

Methods were described sufficiently, however this experiment reflect as monitoring protocol study which strongly suggest authors to include diagram/schematic of experimental set-up. This may allow readers to follow the same experiment protocol for other study.

Detail comment refer to suggestion (Attachment).

Validity of the findings

Result obtained in the study addressed the aim of this study with sufficient statistic prove and highlighted in the conclusion.

Additional comments

This study designed to investigate that fathead minnows require an intact olfactory system to avoid copper exposure. This study is interesting and important as highlighted in the manuscript to fill up the missing knowledge from previous studies. This study found that fathead minnows in control and mock-anosmic groups were able to show avoidance behavior, but anosmic group was not. Thus, this study proven that fish (fathead minnows) require olfactory system to sense copper contamination.

Experiment design was interesting however, the whole manuscript was only presented with ONE (1) figure as result. Of all, experiment protocols authors performed, perhaps there may have more other data that can be presented in this manuscript. In fact, a diagram/schematic of experiment design should be supplied. Perhaps, time response to copper exposure as addition?


Major concern
1. Introduction lack of clear objective, although authors highlighted missing knowledge from previous study. However, lack of hypothesis in this section.
2. How the experiment was performed to maintain and certainly ensure the concentration of the 1.5mg/L copper sulphate at 10 ml administrated into exposure tank at a concentration of 10μg/L? before exposure, is there any recovery test perform to ensure the experiment set up is giving desire concentration?
3. Result on 1 graph, seem like not convince enough. Although the graph reflect the purpose of study with clear indication of fish avoid to copper. However, a whole manuscript with only 1 figure. Is there any data available?
4. Indicate present study was EEGs or EOGs, also use this term in discussion section as well.
5. As this study using video camera to capture behavior and behavioural response was mentioned in M&M section. Other than time spending, is there any other behavioural quantitative data available?
6. L126 – ‘transferred to other protocoals’? what are these protocols? Is there any data available from these protocols?
7. Overall discussion lack to discussing result obtained and explain what are the mechanism leading to this fish avoid and no response toward copper? Explanation about the disturbance about cation, anion and iono sensory functionality will be more interesting and important to highlight the novelty of this study.

Minor concern
L25 – language ‘to find food, identify kin, evade predator and mating’.
L75 – what is the ‘other structures’ provided?
L77 – ‘an inability to right themselves’, what does it mean ‘right themselves? Adapt? Please describe in detail.
L82 – ‘aged to ensure oxygen level …..’ what it mean by ‘aged’
L104-106 – how identification of fish done? This sentence a bit confusing. Is there any schematic or diagram of the experiment set-p available? It is difficult to image how experiment set-up was and difficult for reader to follow or repeat in future. Suggest to supply a schematic experiment design.
L145-146 – should describe further the important and mechanism need by fathead minnows to avoid copper exposure.
L149 – was the present fathead minnows is EOG or EEG? Define
L158 – compare as with present study that used only 10μg/L .. was the concentration play am important factor?
L177-178 – how copper impair olfactory sensitivity?

Annotated reviews are not available for download in order to protect the identity of reviewers who chose to remain anonymous.

Reviewer 3 ·

Basic reporting

A very easy read. Authors get their point across well without confusing the reader.

Experimental design

Experimental design is very clear and has sufficient description to be replicable.

Validity of the findings

I believe behavioral study can be applied for interdisciplinary research in filling knowledge gaps i.e.: aquaculture and fisheries.

Additional comments

Abstract:
1) Please include the concentration of copper used inthis study

Introduction:
2) Lines 26-31: Include a recent study (at least 10 yrs back) for each cation listed.
3) Lines 33-49: lack of citation on same species fathead minnows (Pimephales promelas) to justify baseline copper toxicity used in this study.

Materials & methods:
4) Line 138: "(Morales, 2017)" does not match with reference.

General comments:
5) Hara (1981) has been cited throughout the manuscript. I understand authors might base their current study on this particular 1981 study but are there any similar studies in between?

---

## Round 0.2 · Minor Revisions

I thank the authors for following through with the comments and suggestions of the reviewers. I agree with the reviewer that the current version of the manuscript needs only minor revision before acceptance. I look forward to your revised manuscript.

Reviewer 1 ·

Basic reporting

No comment.

Experimental design

No comment.

Validity of the findings

No comment.

Additional comments

This is the revised version of a manuscript that I previously reviewed. The authors have adequately addressed the issues raised by both reviewers, and I only have very minor comments:

Line 27: Calcium does not always induce avoidance behaviour in fish: the study by Bodznick showed that salmon use it as one odorant (out of a mixture of many) to identify home lake water; that is, the response depended on its concentration.

Line 41: ...some also demonstrate preference or no response. (I am not sure why 'no response' requires quotation marks)

Line 148: Raw data have been provided...('data' is the plural of 'datum').

Line 159: The data demonstrate...(as above).

Line 172: The study by Hara (1981) has only one author; therefore, the use of the pronoun 'they' may be inappropriate. I am not sure of the journal's policy on this is.

Line 201: ...there is no behavioural response. (The authors did not investigate other types of response).

Reviewer 3 ·

Basic reporting

no comment

Experimental design

no comment

Validity of the findings

no comment

Additional comments

The authors have addressed all my concerns. I recommend the manuscript be accepted for publication.

---

## Round 0.3 · accepted · Accept

Many thanks for improving your manuscript according to all the suggestions provided.